# Cooperation in Science and Innovation between Latin America and the European Union

Simone Belli [1,*] and Jenny Morín Nenoff [2]

1    Faculty of Political Science and Sociology, Complutense University of Madrid, 28040 Madrid, Spain
2    German Academic Exchange Service (DAAD), 53175 Bonn, Germany; jenmornen@gmx.de
*    Correspondence: sbelli@ucm.es

**Abstract:** Since the launch of the Strategic Partnership in 1999, the European Union and Latin America and the Caribbean countries have formed a political agenda for bi-regional dialogues. In this study we present a comprehensive analysis of the political and the technical levels of the bi-regional cooperation. The analytical approach that we develop to study bi-regional STI relations is based on a thorough examination of the legal foundations of the EU, which allows for assessment of the possibilities and limits. We identify the LAC dimension of the EU's scientific policies, offering an overview of the challenges and achievements of bi-regional STI cooperation. These are derived from an analysis of limitations in the current cooperation programs. Additionally, the latter is being connected to the discussion of support needs that are raised by the survey participants. We provide a list of suggestions for further instruments and activities, as the main motive is to strengthen and widen the cooperation with concrete actions.

**Keywords:** scientific policies; EU–LAC cooperation; EU policies; LAC policies; innovation





## 1. Introduction

Teamwork in science has become gradually significant, due to research networks and a common idea that strategic societal challenges can only be focused through science collaboration. Joint research actions are another reason, increasing scientific partnership between countries and organisations, and offering the potential for several added-value factors in relation to non-collaborative research [1]. The forthcoming tasks of scientific research are in fruitful partnerships that can profit scientific growth in less-industrialised countries [2]. Bi-regional cooperation is known to be, particularly, significant for countries whose scientific structure and capacity can benefit from building associations with scientists from organisations abroad. For instance, scientists were found to grow team production by nearly 40%, by co-authoring with external allies [3].

At the 2003 OCDE summit, it was stated that collaboration in science and knowledge, in national and international settings, is considered an essential condition to reach the socioeconomic independence of the emerging country, to improve the amount of scientific co-publications. International co-publications are most quoted [4,5], and there is a positive relation between the number of authors per paper and the number of citations received [6]. Sancho et al. [7] reflect that collaboration in science is connected with better quality and scientific relevance; in addition, from here, the propensity of the administrations is to encourage collaboration in research through mutual and bilateral partnerships. Wagner and Leydesdorff [8] show how the cluster shaped by scientifically advanced nations has extended, and new applicants have joined area networks, such as Latin America and Caribbean (LAC) countries. Similarly, political effects and singular agendas can be perceived to have some consequence on connections. In this paper, we present a comprehensive study of both the political and the technical levels of the bi-regional science, technology, and innovation (STI) collaboration among the European Union (EU) and Latin America and the Caribbean (LAC).

In the scientific area, the EU and LAC share a strong aspiration to cooperate at an international level and are, therefore, linked to each other through a long tradition of collaborations and interactions among scientists and institutions [9]. In legal terms, Article 186 of the Treaty on the Functioning of the European Union (TFEU), explicitly, allows the EU to cooperate with less-developed countries and world-wide organizations, in the areas of research and technological growth [10]. However, we have observed that this possibility has only been applied to some LAC countries, such as Argentina, Brazil, Chile, and Mexico, which share the highest number of co-authorship scientific publications with EU institutions [9]. In general, cooperation with LAC countries is not based on that provision and on specific agreements, but is simply the result of "opening" them to the EU framework programme and its projects, which have been set up and implemented on the basis of the inner-looking competence attributed to the EU, by article 182 of the TFEU. In 2012, the European Commission (EC) planned an EU policy for scientific cooperation with third countries, founded on a double method, openness, and targeted actions. The principle of openness, on the one hand, permits EU scientists to participate with their equals international, and, on the other hand, invites scientists and institutions from third countries to take part in the H2020 agenda. Targeted collaboration actions are developed with strategic partners, based on scientific, commercial, and political conditions. Specific strategies and objectives include beginning international challenges, opening access to research infrastructure (RI), and contributing to the EU's international obligations [10].

A knowledge gap exists in the recent literature, about the policies of the cooperation in science and innovation between EU–LAC countries, which should be filled to help experts and researchers make decisions. For this reason, in this paper we offer an innovative contribution compared to the state-of-the-art policies of EU–LAC cooperation, focusing on specific scientific policy programs and analysing their limitations. This text provides an answer to the research question: How effective are the current scientific policies, to promote collaboration between the two regions? In the first part of the paper, we identify the LAC dimension of the EU's scientific policies. In the second part, we offer an overview of the challenges and achievements of bi-regional STI cooperation. These are derived from an analysis of limitations in the current cooperation programs. Additionally, the latter is being connected to the discussion of support needs, which are raised by the survey participants. At the end of the paper, we provide a list of suggestions for further instruments and activities, as the main motive is to strengthen and widen the cooperation with concrete actions.

## 2. Methodology

A hybrid methodology approach combining quantitative and qualitative analysis [11,12] was used, including the analysis of data collection composed by political agreements/statements, EU–LAC calls, and EU–LAC co-publications. Furthermore, we present the results of a survey conducted among representatives of funding agencies (EU-CELAC Interest Group), focusing on their perception of EU–LAC bi-regional STI cooperation.

The analytical approach is characterised by a thorough examination of the legal foundations of the EU that allows for assessment of the possibilities and limits for EU action, taking into account the narrow set of legal competences, which the EU can exercise in its relations with its regional counterparts, attributed to the EU by the Lisbon Treaty.

Our analytical framework differentiates between:

1. Preconditions: Geographical, demographic, economic and historical considerations; political culture; structure of public revenues,
2. Objectives: Articulation between long-term political goals and intermediate or short-term goals,
3. Instruments: (a) legal rules; (b) common activities; (c) budgetary/redistributions/transfers; (d) diplomatic cooperation,

4. Dimensions: (a) external dimension; (b) effective content (cooperation areas and harmonization of policies); (c) strength (law observance and political commitment; (d) dynamism and adaptability.

A total of 33 representatives from funding agencies from 30 countries were invited to participate. The online survey was sent out in February 2019, and at the end of March 2019 we had received 22 complete responses: 10 responses from LAC, including 2 from different Brazilian funding agencies (national/federal), and 11 responses from the EU, including 2 from different German and Spanish agencies each, as well as one response from Turkey. For more information about the survey and the response, we have uploaded all data in this folder: https://drive.google.com/drive/folders/1Ul4bF-T-GCwp19uLeYSGY3lk6i3 hJhEH?usp=sharing (accessed on 13 March 2022).

Three years have passed since the survey responses, but only now can we share these results because the EU–LAC H2020 project was finalized at the end of 2021, so we have fully discussed the results during our workshops during these years. We have the possibility of sharing and analyzing them, collecting the considerations of each member of the research group that has participated in the project.

The survey aims to reveal key motivators, main challenges, and major achievements of bi-regional EU–LAC cooperation in STI. The main objective of the survey is to observe EU–LAC STI cooperation of enhancing international networks, the internationalization of national STI systems, and the political commitment of countries to foster joint research, as well as specific thematic interests.

## 3. Setting the Scene: The LAC Dimension of the EU's Scientific Policies

For international cooperation with non-member states, the EU established the following three core objectives:

- Reinforcing the EU's excellence and attractiveness in research and innovation, as well as its financial and industrial competitiveness;
- Tackling international social challenges;
- Supporting the union's external strategies.

With regard to targeted cooperation activities, different objectives from the EU perspective have to be distinguished. Concerning LAC countries, the differentiation between emerging economies and developing countries is relevant (Table 1).

**Table 1.** EU purposes in scientific collaboration with targeted third countries.

| Targeted Third Countries | Objectives |
|---|---|
| Industrial countries and emerging economies | Increase EU competitiveness to cooperatively face international challenges, through shared advanced solutions. |
| Developing countries | Complement EU external policies and instruments, by building bi-regional affiliations to give to the environmental growth of these areas. |

Furthermore, 'science diplomacy' is emphasized at the EU level, as a soft power tool to facilitate interactions with key countries and regions. So far, EU science counsellors and officers, who are based in key third countries such as LAC countries, act as science diplomats, but the scale of these kinds of actors could be extended to more stakeholders involved in bi-regional cooperation.

### 3.1. The Evolution and Performance of the Joint Initiative for Research and Innovation (JIRI) Framework

Since the adoption of the first bi-annual Action Plan of Madrid in 2010, the following EU–LAC summits in Santiago de Chile 2013, and Brussels 2015, instead of focusing on a limited set of priority cooperation areas, added more points to the original plan, expanding

it to key areas. Unfortunately, broadening the scope of the bi-regional declarations and action plans became a bad remedy for their lack of concreteness and effectiveness. The few tangible results of the EU–LAC bi-regional relations may have contributed to a perception of a half-hearted cooperation and may induce the EU and LAC countries to refocus on bilateral and sub-regional cooperation (such as MERCOSUR and the Pacific Alliance).

With regard to bi-regional STI cooperation, which is the first item listed in the EU–LAC Action Plan, the transition from the EU–LAC Knowledge Area in 2004 to the Common Research Area (CRA) in 2015/2016 narrowed down the scope of STI cooperation from five to three strategic pillars, namely increased mobility, access to research infrastructures, and jointly addressing common challenges (Figure 1).

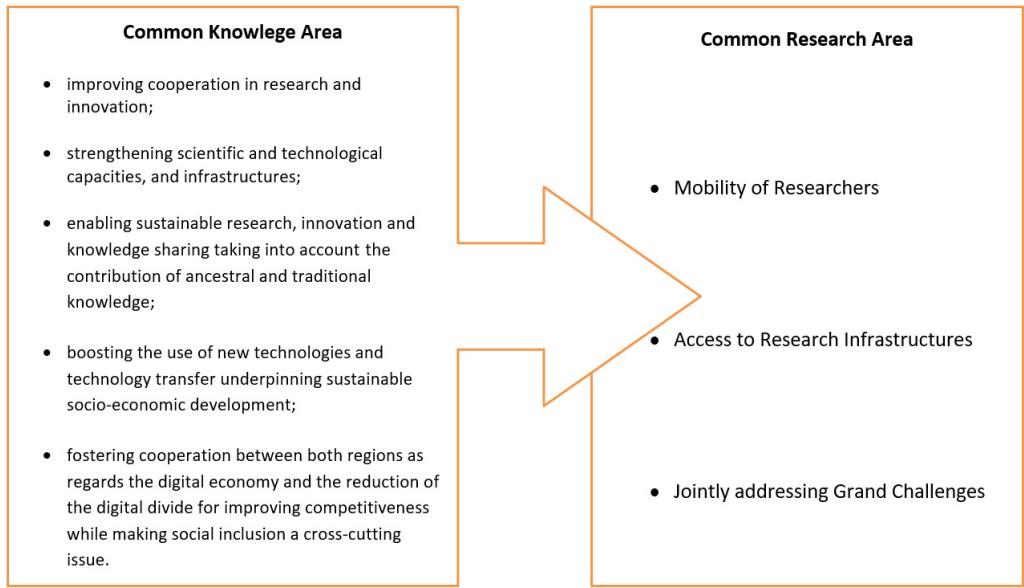

**Figure 1.** From the Common Knowledge Area to the Common Research Area.

However, the key instrument for the implementation of the EU–LAC Knowledge Area, the Joint Initiative for Research and Innovation (JIRI) and its monitoring tool, the roadmap, have not been updated since 2013 and have not been adjusted to the three strategic pillars of the CRA. This lack of evaluation of bi-regional STI cooperation is a shortcoming that, urgently, has to be addressed. At their third senior officials meeting in Brussels in 2013, the representatives of the EU and LAC countries in charge of the implementation of JIRI called for the definition and application of impact and performance indicators, in order to enhance the political visibility of joint activities and achievements.

Thus, the main challenge consists of achieving a demand-driven partnership on equal terms, by overcoming the structural and financial dependence of the JIRI policy dialogue on EC funds. As for most small EU and LAC countries, with lesser developed STI landscapes, it is very difficult to make long-term financial commitments, so a pragmatic variable geometry approach seems to be the most reasonable one.

As the JIRI is a governance structure that lacks a legal entity, it has neither an administration structure of its own, nor a budget at its disposition. Therefore, so far, the JIRI policy dialogue has been technically supported and funded almost exclusively through EU projects, such as ERANet-LAC and ALCUE Net. The resulting leading role of the EC (Directorate-General for Research and Innovation-DG RTD), as the main provider of funds for the bi-regional STI collaboration, led to its dominant agenda-setting position.

The current political situation in LAC, clearly, indicates the diversity of interests and ideas in the region. Due to the lack of a minimum shared denominator for regional integration, Latin America is, obviously, tracking a pragmatic "pick and choose" cooperation approach on a sub-regional level, which is characterised by a diverse set of agreements in a quantity of issue areas, with loser devices of obligation and agreement, which can be

best explained by the concept of "modular regionalism" [13]. In common positions, it can exist stated that in LAC, scientific and innovation plans appreciate as a minor importance, in contrast to knowledge and higher education guidelines, which sum with more formal and economic support [14,15]. While LAC countries with healthier, advanced STI sceneries have placed a better importance on important area-regional innovation structures [14], supporting start-ups (OECD 2016), and emerging innovation value chains in recent years, the inadequate economic support for innovation, mainly from the private area, establishes the main limitation.

### 3.2. Towards the Reduction of the STI Divide in the EU and LAC

The great heterogeneity in LAC, with regard to STI investment patterns, has not changed considerably over time. The fact that Brazil, Mexico, and Argentina account for 91% of regional investment in research is proof of this unchanged status quo. Currently, even these big Latin American countries have reduced public spending for STI, primary due to economic downturns, but, also, because of political scepticism towards the scientific sector [16].

With regard to private investment, the capital stock of innovation is much lower in LAC (13% of GDP) than in EU countries (30% of GDP). Furthermore, only a small fraction of researchers work in the private sector in LAC (24%), compared with the EU average (59%) [16].

In terms of reducing the STI divide, LAC countries should take advantage of the lessons learnt in the EU, where the cohesion policy regarding STI landscapes has, since 2014, been implemented through the combination of economic funds from the European Regional Development Fund (ERDF) and H2020 agenda, as well as other innovation and competitiveness-related union agendas. For the case of LAC, synergies between the Latin America Investment Facility (LAIF) could be explored. Similar to EU rules, the Smart Specialisation Strategy (RIS3) could be established as a requirement, in order to access the fund.

As this RIS3 strategy is founded on the awareness that counties want to shape competitive benefits by corresponding their own research and innovation forces to their own commercial forces, smaller LAC countries could, also, be empowered to recognise a partial set of research and innovation urgencies, plan actions to encourage private research, technology, and growth investment, and place an adequate monitoring and assessment organisation.

Synergies between LAIF and H2020 could be created in a twofold sense. On the one hand, as "upstream action", they could act as a capacity building instrument, and investment in RIs allowing for local-learning mechanisms, fostering personal and institutional capacity to participate in H2020. On the other hand, "downstream actions" should enhance research to business links and, thus, enable the diffusion of the results generated in Horizon 2020 into the market.

Another lesson that STI newcomer LAC countries could learn from EU countries, regarding higher success rates in H2020 calls, are the best practices related to the twinning and teaming instruments, which were created to strengthen the links of research institutions to the TOP20 institutions from EU countries and their research networks. According to a study of the European Parliamentary Research Service (EPRS), 51 percent of the so-far distributed H2020 budget was allocated to the TOP20 institutions, mainly from EU 15 countries, which are the dominant key players with the highest success rates, due to their intense collaboration among each other. Although, the 'spreading of excellence and widening participation' projects are very rare (1% of overall H2020 budget), the 'soft' increase in excellence via joint participation of EU 13 organisations in projects with the TOP20 institutions applies to hundreds of EU 13 teams [17].

*3.3. The Increasing Relevance of Open Innovation and SDGs under the Upcoming Horizon Europe Framework Programme*

As research and innovation networks are considered to be the key to the success of an innovation-led bi-regional cooperation [16], it is good news that the EC and the High Representative of the Union for Foreign Affairs and Security Policy recently declared in their joint communication that "the EU should also seek to encourage cooperation between LAC MSMEs and the Enterprise Europe Network. LAC business clusters and networks could take advantage of the European Cluster Collaboration Platform" (EC/High Representative of the Union for Foreign Affairs and Security Policy 16.04.2019, p. 5).

With regard to the increasing significance of open innovation in the frame of the Horizon Europe programme, the commitment of further sharing "experiences in bridging science and the private sector, transferring technology and ideas from the research base into start-ups and industry, and in promoting smart specialisation and innovation at the regional level" sheds light on the yet underexploited potentials of bi-regional cooperation (EC/High Representative of the Union for Foreign Affairs and Security Policy 16.04.2019, p. 5).

As the open innovation concept falls in the gap between business and academia, several projects in LAC were funded under H2020 and/or supported by DG REGIO, providing groundwork for the Horizon Europe programme to build on. These include initiatives such as EULAC-RIS, the ELAN Network, AL-INVEST, and the EU–LAC INNOVACT Platform, as well as ENRICH in Brazil. However, it has to be mentioned that these projects primary targeted middle-income countries in LAC (Argentina, Brazil, Chile, Colombia, Costa Rica, Mexico and Peru) and should be extended to Central America and the Caribbean [18].

Although the European Union will probably have to cope with budget cuts in the face of Brexit, the budget for Horizon Europe will be increased by EUR 20 billion, to a total amount of EUR 100 billion, compared to the current H2020 (Table 2).

**Table 2.** Comparison between Horizon 2020 vs. Horizon Europe.

| Horizon 2020 (2014–2020) | | Horizon Europe (2021–2027) | |
|---|---|---|---|
| Excellent Science | EUR 24.4 bn | Open Science | EUR 25.8 bn |
| Industrial Leadership | EUR 17 bn | Global Challenges and Industrial Competitiveness | EUR 52.7 bn |
| Societal Challenges | EUR 29.7 bn | Open Innovation | EUR 13.5 bn |

The recent restructuring of the Directorate-General for Research and Innovation (DG RTD) reflects the new pillar structure and gives evidence of the growing importance of effective STI policies, in order to align the achievement of the SDGs with industrial competitiveness. Both objectives constitute the second pillar of Horizon Europe, which accounts for 50% of the overall budget.

As past attempts to stimulate innovations based on the mainstream innovation theory have shown their limits in developing or underperforming regions, for future STI cooperation, it would be advisable to take a more critical and diversified attitude towards innovation concepts and strategies, which include serious socio-economic traditions, political cultures, and regional identities (EULAC Focus WP2-Dl-149 2019). Concepts such as transformative or sustainable innovation hint in this direction, relating science and technology (S&T) policy to social needs and the challenges of sustainable development [16].

Furthermore, such transdisciplinary research topics enhance the inclusion of the social sciences and humanities (SSH) in cooperation projects, which are dominated by research areas related to natural sciences. In this context, the SINCERE project can be labelled as a best practice example, given the fact that one of the key activities performed by the project, in order to achieve the main objective of implementing the JPI Climate Strategic Research and Innovation Agenda 2016–2025 (SRIA), is to promote/integrate the social sciences and

humanities in global (mainstream) climate change research (JPI Climate—Coordination and Support Action (CSA): SINCERE 2018).

## 4. Main Drivers, Challenges, and Achievements of Bi-Regional STI Cooperation

The survey conducted among the members of the Interest Group (IG), which encompasses the representatives of funding agencies who already participated in the ERANet-LAC joint calls, aims at giving a deeper insight into their perception of the bi-regional EU–LAC policy dialogue and joint funding actions in STI.

It reveals that less than 5% of the respondents think that the cooperation is "weak or unsuccessful", indicating that it has been fruitful and beneficial for both the EU and LAC. However, 50% classify the bi-regional cooperation as "half-hearted or inconsistent", which could be attributable to the current suspension of the SOM (Figure 2).

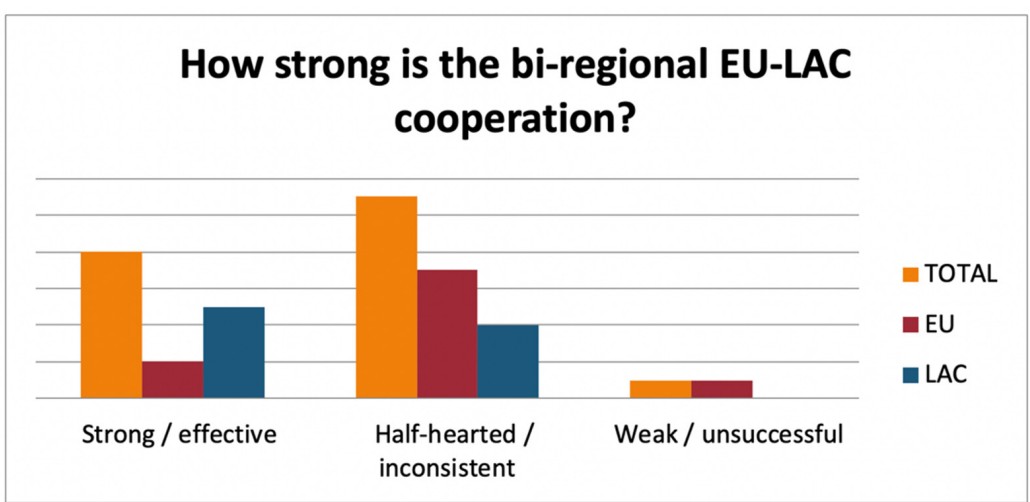

**Figure 2.** Strength of the bi-regional EU–LAC STI cooperation.

Interestingly, the majority of the respondents from LAC believe that the cooperation is strong/effective, followed closely by those who believe it is half-hearted/inconsistent. No respondent from LAC classified the bi-regional cooperation as weak/unsuccessful. The approval rate is lower when considering the participants from the EU, of which the majority believes that the cooperation is half-hearted/inconsistent, and only 17% of the EU respondents believe the cooperation is strong/effective.

As Figure 3 shows the participation of LAC countries in FP7 (2007–2013) and H2020 (2014–2020), some interesting points can be noticed. In the first place, the overall participation shows, at the moment, a decline in overall LAC participation, of approximately 10%.

Not surprisingly, countries with STI agreements with the EU are the ones who take part the most in the Framework Programs. Brazil, Mexico, Argentina, and Chile made up 70% of the LAC participation during FP7 and, currently, stand for 74% in H2020. Brazil, which is the most active country, comprised 30% of the participation during FP7 and, currently, stands for 28% in H2020. However, due to the changes of eligibility criteria for high-income countries (emerging economies), the participation of Brazil and Mexico is declining, while Argentina and Chile show a high increase. Under H2020, Brazil and Mexico are no longer automatically eligible for funding, rather they are responsible for co-funding and must finance the participation of their researchers in H2020 projects.

Nonetheless, the decline in the participation rate is not only happening in Brazil and Mexico. Several smaller LAC countries, especially from Central America and the Caribbean, have had a smaller participation in H2020 when compared to FP7 (Figure 3). Some of these countries are facing migration crisis, e.g., Nicaragua and Honduras, and could be struggling with brain drain. Moreover, these smaller countries could be having problems

in finding partners and joining consortiums, which is a big problem in the EU programmes, given that they might not have contacts and networks in Europe.

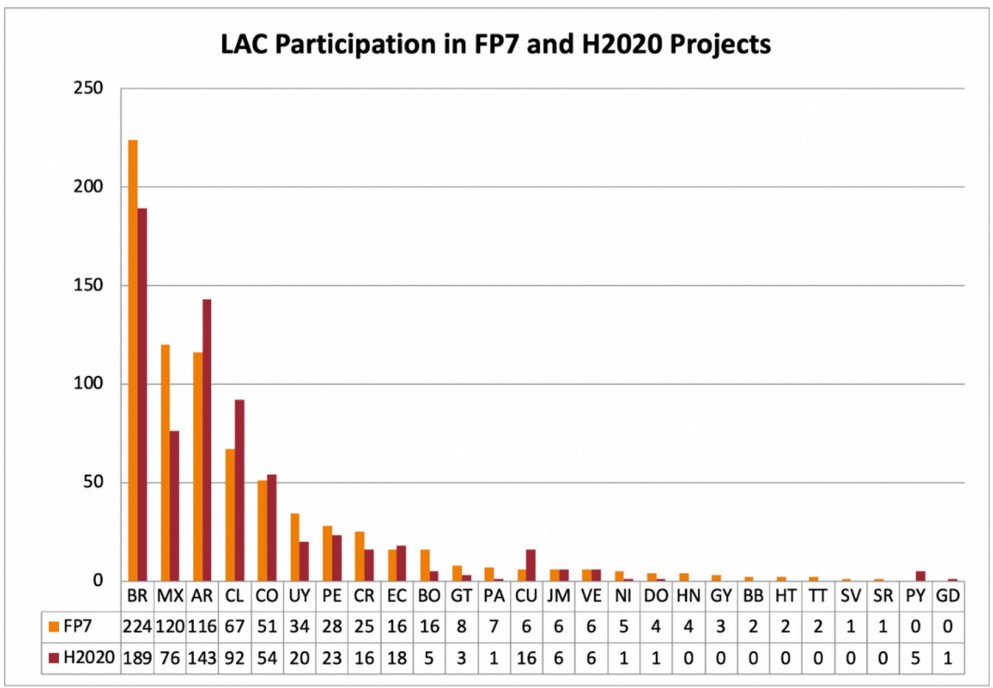

**Figure 3.** LAC participation in FP7 and H2020.

Finally, on a positive aspect, another interesting tendency is that Cuba has increased its participation from 6 projects in FP7 to 16 in H2020. This is, probably, due to the application of the new Political Dialogue and Cooperation Agreement (PDCA) with the EU, which entered into force on 1 November 2017.

Enhancing international networks (81.8%) was named most frequently by participants as a motivator for the collaboration. Internationalization of the national STI system was chosen by 72.7% of participants. Political commitment of countries/regions to foster joint research, as well as specific thematic interests by the partners, were both chosen by more than half (54.6%). Access to funding was less frequently chosen (50%) and special commitment by the participating institutions to develop and participate in joint STI activities was chosen by 40.9%. The least frequently chosen motivator and driver was personal commitment (22.7%).

The lack of financial capacities (72.7%) was given by the majority of the participants from LAC, as well as from the EU, as the main challenge for cooperation. Insufficient support measures provided by national and supranational institutions was the second most frequently chosen challenge (59%). Moreover, 40.9% found problems in the communication flow a challenge.

Lack of commitment of the partners was named as a challenge by 36.4%, mostly by LAC representatives. Preference of other geographical cooperation partners (31.8%) and difficult framework conditions (22.7%) were found to be a minor challenge. Additionally, two participants from EU countries highlighted the following issues: lack of structured capacity building, lack of a "common language" for STI cooperation, and the fact that the regional approach duplicates research themes also present in H2020.

Regarding the lack of financial capacities, according to respondents from big as well as small LAC countries, there is a scarcity of funding, and it was stressed that, in LAC, there is a discontinuity of financial support in long-term activities, and, in the EU, some EU research funding agencies are no longer involved in joint calls of the Interest Group, since they seem to duplicate research themes of H2020 calls.

Concerning the scarcity of funding, it is interesting to remember that in the JIRI document the creation of a "Joint Cooperation Fund" (JCF) was proposed as a new instrument for EU–LAC cooperation under the initiative. Unfortunately, the JCF was never mentioned again. However, at least it is worthwhile to explore the possibility how it could be implemented in order to support long-term cooperation going beyond joint calls (Figure 4).

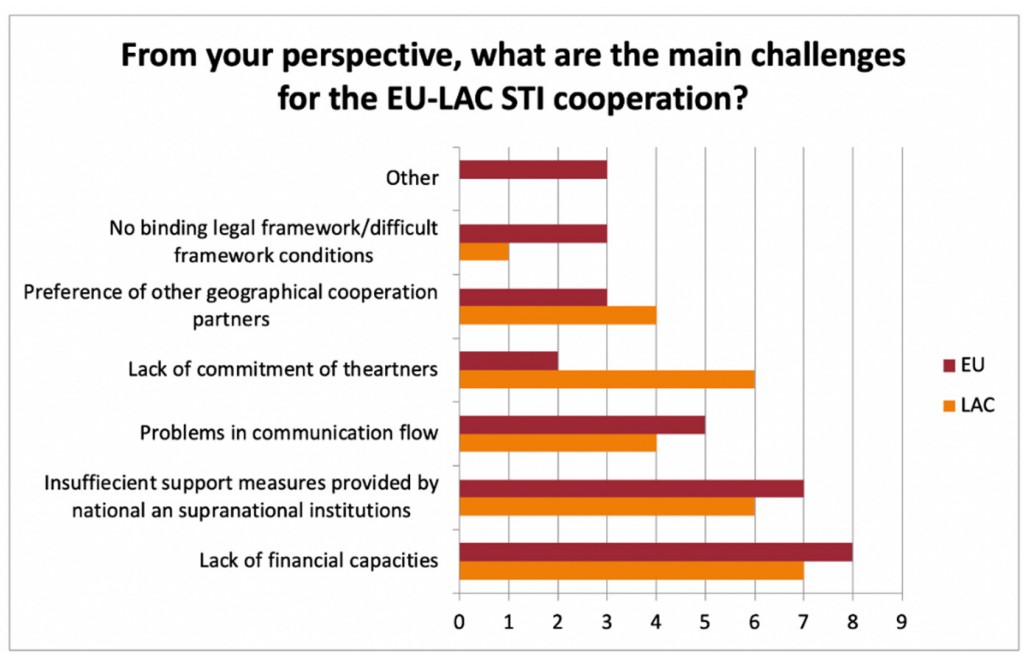

**Figure 4.** Main challenges for EU–LAC STI cooperation.

The effects of the co-financing schemes are ambivalent. While this has led to a more equal partnership between the "LAC big four" and the EU, it is well worth noting that they have not enhanced equality between the "two continents".

Regarding, the second-most-frequently chosen challenge, insufficient support measures provided by national and supranational institutions, survey participants believe that one major deficiency in EU–LAC STI cooperation is a lack of political commitment from both regions, and they, therefore, call for a long-term binding framework and plurennial commitments.

When IG members were asked "what kind of support would the EU–LAC dialogue need to become more effective", in the first place, they stated that there needs to be more political and financial commitment from both sides of the cooperation, with the development of common goals and rules for the cooperation. The need for continuity of the annual policy dialogues was stressed as important for the development of aligned bi-regional strategies. The Interest Group could function as an active interim platform. Additionally, there must be an improvement in discussions and effective communication between both regions as well as between policy makers and other stakeholders. Currently, the EC as the agenda setter and driving force of the bi-regional cooperation seems to be under a certain pressure to fill the EU–LAC Common Research Area (CRA) with life, which is pulling LAC (mainly offering participation in EC initiatives), without systematically asking for needs/demands.

As support measure, the IG members suggest to enhance the technical work, to create a real structure of knowledge that can lead to real priorities. When the ALCUE NET project ended, the technical secretariat for SOM was continued in the Framework of the International Service Facility, without direct and organized involvement from LAC side. Indeed, a greater visibility of the technical secretariat would contribute to a better involvement of all SOM delegates, and, thus, achieve a more balanced relationship vis-á-vis the EC.

Given the fact that it is currently not possible to conduct a dialogue with LAC at a regional level, due to the current political situation of the suspension of the SOM, some IG members consider that bilateral collaboration is more efficient, or, instead, argue that the EC should choose selected strategic partners in LAC as well as involve CYTED and the EU–LAC Foundation, in order to identify a few concrete initiatives creating a benefit for, and high visibility in, both regions, e.g., a Joint Research Centre, which could be similar to Inter-American Institute for Global Change Research (IAI), founded in 1992, with several LAC countries, Canada, and USA.

With regard to the LAC NCP network, one participant from an EU country argues that it is not sustainable and sufficient because it is based on simple training, but does not create a local infrastructure that could be maintained, even if NCPs change jobs. On a different note, it was noted that cooperation should consider the capacities and limitations of the countries and establish SOM working groups involving all the countries, not only the biggest/most influential ones. Plus, it should stimulate the participation of different stakeholders, not only research institutions but also the private sector and entrepreneurs, in order to create innovation value chains [19].

When the Interest Group members were asked in the survey to identify the most efficient instruments in the EU–LAC collaboration, 85% ranked the development of common research priorities and joint calls as the most important instruments. One Interest Group member gives the following argument in the survey, in order to explain the best practice of joint calls: "joint evaluation of proposals, joint decision-making on the basis of the evaluation results, commitment of funding to the best evaluated projects without promoting the national interests at the time of decision-making".

When observing collaboration patterns based on co-publication record numbers, individual countries' activity and collaboration tendencies can be detected. Adding to these observations, the policy-level activity of the country in question can show their role and signal their potential in the future of the collaboration. Argentina, Brazil, Chile, and Mexico are the four LAC countries with STI agreements with the EU. It can be stated that they are the most relevant LAC countries for the bi-regional collaboration, as these four states are, also, the most participative countries in the EU FPs, and the ones that have the most co-publications with the EU [9]. They, also, participated in all SOM meetings, with the exception of Brazil, which was not present in 2012. However, it is observed that the focus is shifting from what has been previously perceived to centralised STI powerhouse nodes to new regions. The international co-authorship network seems to be expanding at the global level, from earlier EU–USA domination. Therefore, the EU is at an advantage to build on its already existing relationship with LAC and explore its potential for further development.

Aside from those LAC countries with bilateral STI agreements, a possible route to benefit both regions can lead through smaller, but in countries that are active in scientific cooperation, on both sides, LAC and the EU. Such countries are, for example, Colombia and Ecuador in the Andean sub-region and Poland in the Central-Eastern European sub-region. Through these countries, a connection can be established between other, as-of-yet inactive smaller countries, which are attractive due to, i.e., interests in specific research topics. Furthermore, smaller countries are very reliant to international cooperation, when it comes to publishing. For them, it is essential to co-publish, meanwhile, Brazil, for instance, produces less than 30% of international collaboration.

A significant reason supporting the necessity of the bi-regional collaboration is the fact that smaller LAC countries rely much more strongly on international collaboration to boost scientific production than their "bigger" neighbours. This phenomenon manifests in some research topics, such as space sciences, where EU–LAC joint collaboration produces much a higher output of publications than either EU28 or LAC does separately. This means, EU–LAC collaboration is very important for space scientists of both regions. It also ties into the tradition of ex-Soviet satellite states' interest in space sciences, where many countries from the EU represent themselves and can find a connection to LAC researchers. These so-called 'smaller' countries, both in the EU and in LAC, show a potential expansion route

for the future of bi-regional collaboration. While EU–LAC collaboration shows the best global growth rate for co-publication registers, and the best growth average per year of all the regions, the well-established and regular LAC STI actors, such as Brazil and Mexico, show, in many fields, stagnating results and, therefore, a significantly decreased motivation for further strengthening the cooperation.

On the other hand, the 'smaller' LAC countries, for instance, show an astonishing evolution of scientific output in recent years [20]. Many of them have produced growth of around 80% in co-publications, whereas Brazil's numbers reveal a less-motivated co-publishing history, with less than 30% of international collaboration, while this rate for other top LAC countries is 50%. The smaller and less developed a country is, the higher the international weight is. Thus, consciously opening to these energetic up-an-coming states might be a way to refresh the currently 'half-heartedly' perceived cooperation.

The majority of participants named joint calls (72.7%) and bi-regional projects (e.g., ALCUE NET, ERANet-LAC, EU–LAC Health) (72.7%) as the most prominent achievements of the collaboration. The following most frequently named achievements are the establishment of the CRA, with its three pillars (54.6%), and regular SOM meetings, as well as the launch of the Interest Group that gives continuity to the successful cooperation in the context of ERANet-LAC. Last but not least, the launch of the LAC National Contactpoints (NCP) network was listed as a major achievement by 40.9% of the participants. As for arguments why, the participants consider these achievements major, they listed reasons such as political framework, the SOM, and the CRA.

The SOM and CRA are both perceived as key parts of a comprehensive approach and their potential—when implemented properly—is recognized. From the point of view of LAC members, the development of regular meetings between high authorities has allowed the establishment of cooperation policies and mechanisms for the involvement and integration of researchers into bi-regional research projects, as well as facilitating trainings through the LAC NCP network. Moreover, regular meetings on strategic issues are recognized to help enhance the capacity of both regions in defining the central targets to be achieved in coming years. As an example, for the importance of SOM as a political instrument, a participant from LAC states that they participate in all these meetings, and the results are the main baseline of their planning of the cooperation with the EU. On the other hand, one participant from an EU country argues that the SOMs are currently not more than a "facade", due to their lack of technical support since the ALCUE NET project, which served as technical secretariat, ended in 2017.

A representative from a small LAC country emphasizes that the CRA facilitates mutual understanding of the major challenges facing the parties involved and prepares a way to work together on common themes. Furthermore, from the LAC perspective, CRA is an ideal scheme for horizontal cooperation.

The bi-regional projects are perceived as key parts of a comprehensive approach. The bi-regional projects, along with the joint calls, have the ability to produce tangible outcomes and face-to-face interaction among the researchers, the active players of the field.

Moreover, a representative from a small LAC country notes that bi-regional projects give LAC countries the space to find partners in European institutions and prepare consortia for H2020 calls. Another partner from Central America perceives bi-regional projects as successful and believes that they have paved the way to much-needed in-depth collaboration in the future. Poland states that there has been great interest among Polish researchers to participate in joint calls for proposals, and those efforts have resulted in bi-regional projects.

The joint calls are seen as a true common effort by both sides. Research-funding agencies state that they actively seek joint funding possibilities with EU and LAC partner institutions, and the joint calls and EU–LAC Interest Group are a way to this.

From the LAC perspective, joint calls provide exposure for institutions in LAC to participate in international research projects, and from the EU perspective they prove that is possible to involve (small) LAC countries in multilateral funding schemes.

On the EU side, the joint calls are said to signal a specific own interest of the institutions involved. This signals the institutions' willingness to invest both time and money (funding of joint projects, personnel). This is a very important basis for a certain degree of sustainability.

The NCP Network is also valued, as a key part of a comprehensive approach. The NCP Network is based on a specific own interest of the institutions involved; thus, the institutions are willing to invest both time and money (funding of joint projects, personnel). This is a very important basis for a sustainable bi-regional partnership. Many survey participants called for the enhancement of the NCP Network.

NCPs operate on a voluntary base and, normally, oversee their NCP tasks on top of their regular duties. If the institutions, which host the respective NCPs, took over more and broader responsibilities with regard to knowledge management, much needed growth in quality could be reached. For instance, a regular problem is the lack of "institutionalized learning"; when an NCP leaves his or her job, the previously established networks and knowledge necessary for the NCP's work is often lost in that location. Therefore, when an NCP is located at a national ministry or council in LAC, this national institution must recognize the added value the NCP brings and take it up to support and, even, financially contribute to the continuity of the NCP's work. In order to achieve the "learning ability" of the NCPs—independently of the actual person behind the position—more effort has to come from the national institutions in LAC.

The NCP network in LAC was implemented with the project support of the EC (in the framework of ALCUE NET and the precursor projects). Uruguay coordinated the network and maintained it to a certain extent after the end of ALCUE NET—recognizing the added benefits of the NCPs network. The individual NCP's work is not financially covered by the EC and has to be financed nationally.

According to the survey results, the Interest Group members affirmed that best practices from other countries or regions (68.2%) are the most powerful instrument that influences their national STI strategies. The following best practices of bi-regional STI cooperation can be identified so far:

a.  ERANet-LAC/EU–LAC Interest Group joint Calls encouraged the EU and smaller LAC countries to participate.
b.  Uruguay, in its function as coordinator in LAC, managed to keep the LAC NCP network alive, after EC funding ended. EC recognized this effort and decided to support again the networking activity in the framework of the International Service Facility.
c.  The EURAXESS Worldwide Brazil node was expanded to the LAC region.
d.  The bi-regional Working Group on Research Infrastructures aims at strengthening the management of LAC RIs and the mutual access to RI in both regions.

## 5. Discussion

### 5.1. The Meaning of Cooperation in Technology and Innovation

Based on the above analysis of limitations in current bi-regional STI cooperation, as well as trends in the context of the next framework program Horizon Europe, we formulate, according to the EU–LAC Focus analytical framework, the following preconditions and political (long-term) goals, as well as short-term goals, for future bi-regional STI cooperation.

In formulating the preconditions and objectives for future bi-regional EU–LAC STI cooperation, it is important to bear in mind the following structural inter- and intra-regional differences of STI policies.

- Acknowledgement of modular regionalism in LAC, which is characterized by a variable geometry approach of cooperation between sub-regions and sub-regional projects in a number of issue-areas, with loser mechanisms of commitment and compliance;
- Common understanding of scientific values: rationality, transparency, and universality;
- Mutual acceptance of diverse understandings of research excellence and innovation strategies;
- Awareness of different structures of STI landscapes regarding the balance of basic and applied research;

- Awareness of different structures and amounts of public and private funds for long-term investment in R&I;
- Complementarity of STI cooperation with mutual trade interests and/or development cooperation.

Having defined these preconditions, we describe the main objectives of future bi-regional STI cooperation:

a. Strengthen bi-regional dialogue to tackle global challenges and contribute to SDGs;
b. Reduce the STI divide in LAC and the EU, by intensifying the collaboration of low-performing countries with the TOP20 institutions/networks under H2020;
c. Foster innovation cooperation by reducing the digital divide and strengthen technology transfer.

We consider these three objectives as big political goal to achieve in the EU–LAC relations, for a long-term strategy. For this, that we introduce five specific objectives for a short-term strategy:

a. Agree on a bi-regional funding mechanism to expand the scope of action of the EU–LAC foundation, in order to follow up the JIRI evaluation and monitoring tool;
b. Create synergies between JIRI and the R&I-related trans-regional programmes of the Partnership Instrument (PI), such as, e.g., the International Urban Cooperation (IUC) and the International Digital Cooperation;
c. Promote country-specific and goal-related excellence and innovation strategies;
d. Set up joint industrial Ph.D. programs and promote mutual learning of creating technology transfer offices at universities;
e. Develop special twinning and teaming instruments (similar to those for EU13 countries) for lesser-developed LAC countries. in order to introduce their research institutions to the TOP20 institutions and their research networks.

To achieve these objectives, it is important to have specific instruments and tools. According to the typology of exclusivity and specificity of international arrangements for STI cooperation by Kaiser [21], the International Cooperation-Nets, as well as bilateral STI agreements among nation states, are grouped under arrangements of low exclusivity and low specificity, suggesting that STI policy actors believe that cooperation itself is advantageous, even if there is no clear understanding of the potential benefits. However, they suggest that arrangements of low exclusivity and high specificity are the most likely context for international STI cooperation, since the involved partners seek to balance their gains and burdens, either targeting certain common (global) challenges or basic research and long-term-knowledge production, without an immediate impact on the country's competitiveness. If STI policy actors aim at improving their own competitive position in a scientific topic, with an achieved comparative advantage and/or strategic importance, they will establish a highly exclusive and specific cooperation arrangement [21].

The development of the participation of funding agencies from European MS/AC is not that constant. With an increase in participation from the first to the second call, in the third call, the number decreased again.

Several countries that joined the EU in the enlargement in 2004 are participating in the joint calls, although they are not represented in the SOM meetings. These include Romania in the first and second calls, Latvia in the second call, and Poland in the second and third calls.

Most interestingly, in countries with less experience in cooperation with the other region, there has been an increase in successful projects with their participation over time. Romania increased from two to six projects. The same happened for Panama, which did not fund any projects in the first call, but in the second call researchers from their country are already involved with four projects. Moreover, Poland entered the joint activities in the second call, with eight successful projects with Polish participation.

This new platform has to be linked to and make use of existing platforms, which were established, e.g., in previous EU-funded projects such as ALCUE NET and ERANet-LAC.

*5.2. The Cooperation between Two Continents and Open Innovation*

For these reasons, we suggest these six actions to improve bi-regional STI cooperation:

a.　　Increase visibility and promote open access platforms

To use the EC International Cooperation for Research and Innovation webpage as a comprehensive repository of bi-regional cooperation, and link it to the EU–LAC platform, in order to improve the presentation of international networks and funding opportunities relevant for LAC stakeholders [22]. Promote LAC (non-market-based) open-access platforms, such as SCIELO, in order to attract EU researchers who normally use (market-based) open-access platforms, such as ResearchGate.net and Academia.edu, to push their academic careers. Develop a MOOC, which gives a good insight into the best practices of bi-regional cooperation (such as the best practices presented by the alliance of universities Grupo Montevideo).

b.　　Strengthen LAC as unique research destination

Develop a standardized concept and guidelines for an awareness/marketing campaign (Burroni, Gherardini and Scalise, 2019), in order to strengthen LAC as a research destination for EU Ph.D./Postdoc fellows. Support the activities of the SOM Working Group on Research Infrastructures regarding the development of a common understanding of research infrastructures.

c.　　Enhance mobility of researchers in both directions

Further enhance participation of LAC students and scientists in Marie Skłodowska-Curie Actions (MSCA) and European Research Council (ERC) mobility schemes by intensifying synergies/exchange of information of EURAXESS worldwide with the NCP–LAC network. Further promote industrial Ph.D. programmes in LAC, in order to strengthen the link of research and innovation by developing personal networks in which knowledge between companies and universities can be disseminated [15]. Establish a close interaction between the EURAXESS worldwide team and the mobility platform *Campus Iberoamérica* as well as (sub)regional networks/alliances in LAC, in order to support the implementation of their internationalisation strategies/efforts. Aligning and harmonise bi-regional, regional, and national efforts to avoid unintended side-effects, for example, duplications or undercutting of scholarship programmes. Addressing the gender gap [23] by integrating the Responsible Research and Innovation (RRI) approach in the MSCA and ERC mobility schemes, by setting up a special programme for (young) female early-career researchers, which takes into account the special needs of the family members (care costs, school fees, dual-career offers for (academic) spouses). Set up a study that explores the obstacles that hinder female LAC researchers to benefit from longer research fellowships abroad. Find out about possible options to tailor the existing mobility schemes (MSCA, ERC) to the specific needs of female researchers, taking into account their cultural and regional background [8].

d.　　Strengthen joint research funding

Building on the best practice of the ERANet-LAC/Interest Group joint research funding activities, maintaining the variable geometry approach, by matching countries with well-developed STI landscapes and those with a moderate/low STI performance, sticking to the diversity of national funding regulations [24,25]. Set up a partnering request, according to the new partnership policy approach under Horizon Europe (FP 9), in the frame of the Service Facility Contract: a new obligatory requirement should be the involvement of at least one partner organisation from the EU 13 countries in order to better integrate them into the network of the TOP20 organisations from EU15 countries. Increasing the impact of the instrument "spreading excellence, widening participation", by helping research institutions from EU countries to improve the quality of their administrative skills, which increases eligibility and, therefore, also, their success rate and involvement in calls targeted to the LAC region [9,26,27].

Further promote involvement of LAC in the thematic Joint Programming Initiatives on Climate, Agriculture, Food Security, Climate Change (FACCE-JPI), Water, and Oceans

by, also, integrating the Social Sciences and Humanities in the mainstream natural science driven global climate change research and inform on the best practices on the EC International Cooperation for the Research and Innovation webpage (Action 1). Further promote participation of LAC institutions in international funding programs such as the Belmont Forum and explore synergies with Interest Group joint research funding (Cortes-Sanchez, 2019).

e.   Establish synergies to foster industry–society–academia cooperation

Learning from good practices reducing the R&I divide in the EU by funding synergies from ERDF—European Regional Development Fund and H2020, enhancing, on the one hand, "upstream actions" focusing on infrastructure investment and capacity building, and, on the other hand, "downstream actions" enabling the diffusion into the market of research results from Horizon 2020 projects. Exploring funding synergies between the Blending Facility (DG DevCo) and FP 9 Horizon Europe, in order to reduce the R&D divide and foster the research to business (R2B) links in LAC [17,28,29].

- The 18 LAC countries: The Latin America Investment Facility (LAIF), which is a Development Cooperation Instrument (DCI) could be invested in STI infrastructure (Investment grants), as well as in diffusion to market programs (risk capital), creating synergies with the FP9 Horizon Europe research funding.
- The 15 Caribbean countries (ACP): The Caribbean Investment Facility (CIF), which is an instrument of the European Development Fund (EDF) could be invested in STI infrastructure (Investment grants), as well as in diffusion to market programs (Risk capital) creating synergies with the FP9 Horizon Europe research funding. Using the policy advice of the Caribbean Council on Science & Technology (CCST) can be supported on its way to be transformed into an STI observatory, in order to have a solid database, to generate a Caribbean Regional System of Innovation.

Opening the SME instrument to LAC enterprises with certain conditionalities in place, securing the economic interest of the EU MS and SMEs by launching specific calls, in order to attract well-developed key technologies in LAC to cooperate with EU businesses [3,30].

f.   Align higher education with regional research and innovation policies

Point out synergies and exploit the cooperation potential of different schemes such as Erasmus, Framework programmes, COST actions, and ALFA actions promoted by several institutions (DG RTD, DG EAC, DG Regio, OECD, OEI), by extending the mandate/resources of existing platforms (EU–LAC Foundation, Obreal), or by creating new ones (e.g., through a dedicated action in the new Framework programme). Strengthen interdependencies between higher education (HE) and regional innovation policies, such as the RIS3 strategies on smart specialisation [31,32]. Adding a perspective to RRI policies helps to make HE systems more receptive to societal needs and, thereby, maximises the potential impact of cooperation [33,34]. Promoting the exchange of good practices of successful implemented smart specialization strategies (S3 platform worldwide), support matchmakings between LAC and the EU, and promote international sustainable innovation value chains.

## 6. Conclusions

In the paper, we have filled the knowledge gap about the policies of the cooperation in Science and Innovation between EU–LAC countries, to help experts and researchers to make decisions. We have offered an innovative contribution compared to the state-of-the-art policies in EU–LAC cooperation, focusing on specific scientific policy programs and analysing their limitations. This text has answered the initial research question, analyzing the effectiveness of the current scientific policies, showing the positive and negative aspects of each one. We have identified the LAC dimension of the EU's scientific policies, offering an overview of the challenges and achievements of bi-regional STI cooperation. These are derived from an analysis of limitations in the current cooperation programs.

One of the main limitations of this work is methodological. We believe that the dataset would be improved, by using a survey more geared towards collecting information from specific LAC programs, since, in this work, we have focused our attention from a collaborative community perspective and less from a non-European perspective. For this reason, we suggest that future researchers follow this line, in order to have a more specific mapping of the entire bi-regional context of EU–LAC scientific cooperation.

**Author Contributions:** Conceptualization, J.M.N.; methodology, J.M.N. and S.B.; formal analysis, S.B. and J.M.N.; investigation, S.B. and J.M.N.; writing—original draft preparation, J.M.N.; writing—review and editing, S.B.; visualization, J.M.N.; funding acquisition, S.B. and J.M.N. All authors have read and agreed to the published version of the manuscript.

**Funding:** This project has been developed within the framework of the research and innovation project. European Union's Horizon 2020 No 693781—"Giving focus to the Cultural, Scientific and Social Dimension of EU—CELAC Relations". Comunidad de Madrid No 2018-T1/SOC-10409 "Atracción de Talento (Modalidad 1)".

**Institutional Review Board Statement:** Not applicable.

**Informed Consent Statement:** Not applicable.

**Data Availability Statement:** The data reported could be found at: https://drive.google.com/drive/folders/1Ul4bF-T-GCwp19uLeYSGY3lk6i3hJhEH?usp=sharing (accessed on 13 March 2022).

**Acknowledgments:** Authors would like to thank Eszter Simon, Sophie von Knebel, Wolfgang Haider, and Ana Andrade Good God for collaborating in this research.

**Conflicts of Interest:** The authors declare no conflict of interest.

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
