# Peer review of "Cooperation in Science and Innovation between Latin America and the European Union"

_2199-8531, doi:10.3390/joitmc8020094_

Round 1
Reviewer 1 Report
This manuscript has potential and may be suitable for publication in this journal. However, some improvements are needed, which I describe below.
- The title of this article is very convoluted and is in danger of not drawing the attention of potential readers of the journal. I recommend that the authors improve it.
- The Abstract does not provide an exhaustive overview of the study, please refer to the journal guidelines ie: 1) Background: Place the question addressed in a broad context and highlight the purpose of the study; 2) Methods: Describe briefly the main methods or treatments applied. Include any relevant preregistration numbers, and species and strains of any animals used. 3) Results: Summarize the article's main findings; and 4) Conclusion: Indicate the main conclusions or interpretations.
- In the Introduction paragraph, the authors should provide readers with the justification for this study, identifying knowledge gaps in the recent literature that should be filled. In addition, the authors should explain in more detail to the readers what is the innovative contribution of their research compared to the state-of-the-art.
- Also in the Introduction paragraph, a more detailed analysis of the literature would be useful.
- In the Methodology paragraph the authors refer to a hybrid qualitative/quantitative approach, it is necessary that the authors refer to recent literature to justify this methodological choice.
- Additionally, it is appropriate to justify why the survey references 2019. Three years have passed, can the result obtained still be considered current? The authors need to discuss this question in detail.
- For the sake of clarity, authors should present readers with the detailed structure of the survey they applied. The current description is too concise.
Author Response
Dear Reviewer,
Thank you for these important suggestions.
I have introduced all of them to the current version of the manuscript. Also I have uploaded a version with the changes introduced.
All the best,

Reviewer 2 Report
Dear Author(s),
Thank you for submitting this manuscript.
Although the paper addresses in an original way an interesting topic, in my opinion in its current form it lacks some methodological rigor.
My main concern is about the structure of the work: after the introduction (where one or more research questions should be clearly stated) and before the methodology, I would include a section dedicated to the theoretical framework of the paper (also stating why your work is original with respect to what is already known on the topic); a section specifically dedicated to findings (where to describe results without interpretation) should be followed by discussions (where to interpret findings); finally, a concluding section, where to clearly answer to the research question(s) raised in the introduction, as well as limitations and practical/policy implications, is needed.
Moreover, the abstract should be extended, to include some more details on the methodology and main findings and implications of your work.
Best wishes!
Author Response

(The authors gave the same response as above.)

Round 2
Reviewer 1 Report
Dear Authors,
I have appreciated the improvements you have made to your manuscript by following the reviewers' suggestions.
Therefore, I consider this latest version of the article suitable for publication in the journal.
Author Response
Dear Reviewer,
Thank you for your decision.
All the best
Reviewer 2 Report
Dear Author(s),
The paper has consistently improved with respect to the previous version. However, a "Conclusions" section, where to address answers to research questions, as well as possibile limitations and policy and practical implications of your manuscript, is still missing. Please include it in your revised work.
Best wishes.
Author Response
Dear Reviewer,
Thank you for this suggestion. We have added the conclusions section.
All the best,
Round 3
Reviewer 2 Report
Dear Author(s),
The manuscript can now be accepted in its current form.
Thank you!
Author Response
Dear Reviewer,
Thank you for accepting this manuscript.
all the best,